# "Factors associated with provider unwillingness to perform induced abortion in Argentina: A cross-sectional study in four provinces following the legalization of abortion on request"

Paula Vázquez[1]*, Carolina Nigri[1], Verónica Pingray[1], Luz Gibbons[1], Sandra Formia[1], Analía Messina[2], Claudia Castro[3], Cintia Jacobi[4], Adriana Martiarena[5], Susana Velazco[6], Ana Langer[7], Jewel Gausman[7], R. Rima Jolivet[7], Caitlin R. Williams[1], Mabel Berrueta[1]

1 Institute for Clinical Effectiveness and Health Policy (Instituto de Efectividad Clínica y Sanitaria (IECS)), Buenos Aires, Argentina, 2 Servicio de Obstetricia, Hospital General de Agudos Dr. T. Álvarez, Buenos Aires, Argentina, 3 Dirección Provincial de Maternidad e Infancia, Ministerio de Salud, San Salvador de Jujuy, Provincia de Jujuy, Argentina, 4 Dirección de Maternidad Infancia y Adolescencia, Subsecretaría de Salud, Santa Rosa, Provincia de La Pampa, Argentina, 5 Programa Materno Infantil, Ministerio de Salud, Región Sanitaria V, San Isidro, Provincia de Buenos Aires, Argentina, 6 Dirección de Maternidad e Infancia, Secretaría de Servicios de Salud, Ministerio de Salud Pública, Salta, Provincia de Salta, Argentina, 7 Women and Health Initiative, Department of Global Health and Population, Harvard University T.H. Chan School of Public Health, Boston, Massachusetts, United States of America

* pvazquez@iecs.org.ar

## Abstract

### Background

The 2020 Law on Access to the Voluntary Interruption of Pregnancy is a landmark piece of legislation regarding access to abortion in Argentina. Under the new law, abortion is legal up to 14 weeks and 6 days gestation, with exceptions made to the gestational age limit to save a woman´s life, to preserve a woman´s health, and in case of rape. However, widespread refusal to provide care by authorized health providers (due to conscientious objection or lack of awareness of the new law) could hinder access to legal abortion. This study aimed to assess knowledge of the current legal framework and willingness to perform abortions by authorized professionals in Argentina, to compare perceptions about any requirements necessary to perform abortions on legal grounds between willing and unwilling providers and to explore factors associated with refusal to provide care.

### Methods

We conducted a cross-sectional study based on a self-administered, anonymous survey to authorized abortion providers in public health facilities in four provinces of Argentina.

**Data Availability Statement:** All data have been anonymized to ensure compliance with human subject protections and study protocols. The

anonymized data underlying the findings are deposited here: Jolivet, Rima; Gausman, Jewel; Adanu, Richard; Bandoh, Delia; Berrueta, Mabel; Chakraborty, Suchandrima; Kenu, Ernest; Khan, Nizamuddin; Odikro, Magdalene; Pingray, Veronica; Ramesh, Sowmya; Vázquez, Paula; Williams, Caitlin; Langer, Ana, 2022, "Validation data for measuring the "Legal Status of Abortion"", https://doi.org/10.7910/DVN/OCOE3B, Harvard Dataverse, V1, UNF:6:S77IPSgJW3AHbZ/gVeX/UA== [fileUNF]. This work is licensed under a Creative Commons Attribution 4.0 International License.

**Funding:** This work was supported by the Bill and Melinda Gates Foundation: https://www.gatesfoundation.org/ RRJ and AL received the award for Improving Maternal Health Measurement (IMHM) Capacity and Use through which this work was funded, with grant number OPP1169546 The funders had no role in study design, data collection and analysis, decision to publish, or preparation of the manuscript.

**Competing interests:** The authors have declared that no competing interests exist.

## Findings

Most authorized providers knew the grounds upon which it is currently legal to perform abortions; however, almost half reported being unwilling to perform abortions, mainly due to conscientious objection. Both willing and unwilling providers believed there were additional requirements not actually stipulated by law. Using logistic regression, we found that province where providers serve, working in a tertiary level facility, and older age were factors associated with unwillingness to provide care.

## Conclusions

The results of our study indicate that, even in a favorable legal context, barriers at the provider level may hinder access to abortion in Argentina. They help to demonstrate the need for specific actions that can improve access such as training, further research and public policies that guarantee facilities have sufficient professionals willing to provide abortion care.

## Introduction/Background

Abortion was broadly legalized in Argentina in December 2020 after an unprecedented social mobilization [1]. The new law passed was the result of a long process that began to take shape many years earlier, led by feminist activists, networks of abortion providers, and community groups that helped people self-manage abortions [2]. Feminist movements inspired women of different generations to march massively through the streets of the National Congress wearing green insignias (the color that represents support for legal and safe abortion) creating a "green wave". These mass movements were complemented by an extensive and broadcasted debate of the bill on Access to the Voluntary Interruption of Pregnancy in the Parliament, to produce a social and cultural change that gave support to legalization [3, 4].

The law is expected to improve public health, as laws legalizing abortion create enabling environments for skilled providers and the access to safe methods [5]. For example, in South Africa and Nepal, maternal mortality declined after the liberalization of abortion law [6]. In addition, legalization implies recognition on the part of the State its of obligations towards pregnant people as rights-bearers vis-à-vis the right to safe abortion and post-abortion care [7]. The passage of the new law in Argentina is consistent with international human rights treaties ratified by the country and underscores the commitment of the Argentine state to public health and sexual and reproductive rights, guaranteeing the safety and timeliness of abortions for pregnant people [8, 9]. Legalization of abortion in Argentina is also an important achievement that reflects broader societal change across Latin America and the Caribbean, where access to abortion has increased substantially in the past decade [10].

Under the new law, pregnant people can access abortion up until 14 weeks and 6 days of gestation without need for any justification (i.e., on request). Abortion is also legal (without gestational age restrictions) to save a woman´s life, preserve a woman´s health, and in case of rape [8]. However, even in favorable legal contexts, barriers imposed at provider and/or facility level may hinder access to abortion. Poor knowledge of legal frameworks, misinterpretation of laws and protocols that are not clearly written, and invocation of conscientious objection are common problems that have been detected in Argentina and worldwide [11–14].

Poor knowledge of the legal status of abortion may cause providers to mistakenly deny care. For example, a provider who erroneously believes that abortion upon request is not legal may refuse to provide care even if they would personally be willing to perform the abortion. This issue could be magnified in settings where new laws have recently been implemented. It is therefore crucial that any provider authorized to perform abortion, have the correct knowledge of abortion laws and policies to ensure access to the practice and also to protect sexual and reproductive rights [12, 15]. In Argentina all physicians, are legally authorized to perform both surgical and medical induced abortions [16].

In addition, providers may believe that there are additional restrictions or requirements beyond those codified in domestic laws. In Argentina, for example, under the new law, no additional restrictions (beyond gestational age for abortion "on request") are required. Examples of ignorance about restrictions might include requesting an official incident report from legal authorities in order to access abortion care for any pregnancy resulting from rape even though one is not required [13], or soliciting informed consent from both the woman and her partner in settings where only a woman´s consent is necessary [17]. Such misinterpretations may contribute to providers denying or delaying access to abortion in Argentina and elsewhere.

However, there are also circumstances in which providers may understand the law, but still refuse to provide care, invoking religious or moral reasons (conscientious objection).

Conscientious objection is manifested when a health care provider refuses to provide abortion services or information on the basis of their personal moral or religious beliefs [18]. It is a phenomenon that has expanded globally and is by far the most common reason for refusal to provide abortion care and other reproductive health services such as contraception and sterilization [19]. Notably, conscientious objection in Argentina is an individual right only, and cannot be exercised at the institutional level. In contrast with how conscientious objection is defined in other countries, in Argentina it can only be invoked to justify refusal to perform abortions and cannot be extended to other related (comprehensive health care) services (e.g., counseling or provision of information on abortion, ultrasound, clinical health care practices, among others). The right to conscientious objection is revoked in cases in which the woman's life is at risk and no other professional is available to perform the practice. In Argentina, certain conditions frame the right of objection: providers must be consistent in their conscientious objector status in both the public and private sectors, and conscientious objection must co-exist with the principles of non-obstruction and good faith [8, 9]. Such limitations were designed to ensure that providers' exercise of conscientious objection did not unduly restrict access to legal abortion. Yet, evidence suggests that conscientious objection has resulted in delays and outright denial of care for women seeking abortions, and has created hostile work environments for providers willing to provide care [20–22]. Despite emerging evidence, it is still unclear how prevalent conscientious objection is and whether there are provider/facility level factors associated with the refusal to provide care. This study seeks to address those two gaps.

In summary, lack of knowledge about laws, incorrect interpretations of regulations, and conscientious objection may constitute important provider-level barriers to abortion access. There are also questions about the ways in which these three issues may be interrelated. This study aimed to assess knowledge of the current legal framework and willingness to perform abortions by authorized professionals and to compare whether there are differences in beliefs about additional requirements and restrictions between those willing and unwilling to provide abortion care in Argentina. We also explored whether there are demographic factors associated with refusal to provide abortion services in the context of the recent removal of legal restrictions.

Recognizing that not everyone who may need abortion care may identify as a woman, in this study we use the terms "women" and "pregnant people" interchangeably to refer to people with reproductive capacity for pregnancy.

## Materials and methods

This cross-sectional study is based on a self-administered and anonymous survey of professionals authorized to perform induced abortions in public health facilities located in four provinces of Argentina. This paper describes a secondary analysis of data collected for validation of the global indicator, "Legal status of abortion" as part of the "Improving Maternal Health Measurement Capacity and Use" research study that aimed to validate 10 indicators from the Ending Preventing Maternal Mortality (EPMM) monitoring framework in three countries: Argentina, Ghana and India [23].

### Participants and sampling

**Settings.** Provinces/districts were selected according to a purposive sampling plan based on a composite index that was described in the research protocol for the larger study [23]. Then, public facilities offering abortion services were selected from 20 randomly selected primary sampling units (PSU), following the same standardized multistage sampling plan used for the Demographic and Health Survey [24]. Finally, as a result of this sampling, five facilities were selected in each province to represent the three levels of care: one tertiary care facility, one secondary care facility, and three primary care facilities. Given that the same sample of facilities selected served in the validation of other indicators of the master project, facilities not employing midwives were excluded.

**Participants.** Eligible participants, were the cadres legally authorized to perform abortion practices (both, medical and surgical) in participating health facilities (specialists in Ob/Gyn and General Practitioners) that provided services of sexual and reproductive health. Facilities' managers provided a list of eligible participants based on the payroll. They were invited to participated, those who gave their consent to participate were included in the study. Exclusion criteria included providers who were unwilling to provide consent. No other exclusion criteria were applied.

### Data collection and management

The survey was piloted with a group of obstetricians for content and cognitive assessment and was then refined. Surveys were conducted from July to October 2021 and were self-administered both electronically and on paper. Eligible professionals received an email with an explanation of the purpose of the survey and an invitation to participate by signing an electronic informed consent form in a secure web-based portal protected by password (REDCap version 11.2.2). Those professionals who gave their consent received another link to complete an anonymous survey within the same portal. A weekly reminder was automatically sent by e-mail from the data center to participants who had not responded to the consent form and/or completed the survey during the study period. Those who preferred to answer the survey on paper gave informed consent and then completed the questionnaire in a private room. Hard copies with a unique, anonymized identification number and containing no personal information were placed in sealed envelopes which were transferred by a certified private courier to the data center in Buenos Aires. De-identified paper surveys were then entered into the data portal. Data anonymity and confidentiality were ensured at each stage of data collection and data analysis. During the recruitment process, the data manager who oversaw the administration of the survey did not have access to the list of providers' names or their responses to the survey.

The staff responsible for entering the paper survey data into the study platform did not have access to the list of provider names or any identifiable information. To avoid possible identification of participating providers, province names were masked through assignment of a random number from 1 to 4. However, we will mention some relevant indicators of the selected provinces (recorded for the year 2017, the latest data available at the time of initiating the selection of the provinces participating in the study) that would allow contextualizing the findings of this study. Of the four provinces selected, two belong to the Northwest region (province 2 and 4) and two to the Central region of Argentina (province 1 and 3). Maternal mortality ratio, was highest in provinces 2 and 4, (34 and 48 maternal deaths per 100,000 live births, respectively) almost double the values for province 1 and for province 3 (27 per 100,000 live births and 20 per 100,000 live births, respectively) and exceeding the national average of 29 per 100,000 live births [25]. In addition, the provinces representing the northwest had a total fertility rate that exceeded the national average value of 2.1, (ranging from 2.5 to 2.7), while in the provinces representing the Central Region the total fertility rate was lower than the national average [26].

The structured questionnaires collected demographic data, knowledge about grounds on which abortion might be legal (to save a woman's life; to preserve a woman's health; in cases of intellectual or cognitive disability of the woman; in cases of rape, in case or incest; in cases of fetal anomaly or impairment; for economic or social reasons; and upon a woman's request) [27] as well as additional restrictions to abortion on legal grounds, and provider attitudes related to the provision of abortion services. In the survey, the term conscientious objection was defined to the participants as *"personal religious or moral reason"* for which professionals may opt out to perform the practice. From the total list of the possible legal grounds for abortion mentioned, for this secondary analysis, we selected only those that are currently legal under the new legislation although providers' knowledge was tested using all possible legal grounds as prompts in the master research project validating the indicator, "legal status of abortion" [23].

## Statistical analysis

First, a descriptive analysis of the providers' characteristics was performed. Next, we calculated the absolute and relative frequencies of providers' knowledge of the four legal grounds for abortion in Argentina (to save a woman's life; to preserve a woman's health; in cases of rape and on request), their willingness to perform a legal abortion on each ground, and the reasons they stated for not providing abortions on legal grounds.

Among providers who correctly identified the legal grounds, their perceptions of the existence of restrictions or additional requirements needed to access legal abortion were explored. To determine whether there were systematic differences between providers willing and unwilling to perform legal abortions, we stratified responses by willingness to provide care. A proportional Z-test was used to compare the results. When the number of cases was too small, a Fisher's exact test was used instead.

To evaluate factors associated with unwillingness to provide abortion care, we first reported the proportion of providers unwilling to provide care by ground. We also performed bivariate and multivariate analyses in which we used Firth's bias-reduced logistic regression model to account for the small number of providers surveyed. This method provides a bias-reduction of the maximum likelihood estimation for small simple sizes as well as yielding finite and consistent estimates [28]. The crude and the adjusted odds ratio with the 95% confidence intervals were reported. In the multivariate analysis, the variables that were associated with the outcome and remained significant after their inclusion in the model were kept. Each of the legal

grounds was analyzed separately. For each legal ground, the distribution of socio-demographic characteristics of the providers who were willing and unwilling to perform abortions were described. To preserve confidentiality, findings were presented with the provinces' names masked using numbers. Statistical analysis was conducted using R software.

## Ethical considerations

This study is a secondary analysis of data collected as part of the "Improving Maternal Health Measurement Capacity and Use" study, which received IRB approval from the ethical review board at the Office of Human Research Administration at Harvard University (IRB19-1086). In Argentina, the institutional ethical review boards of each participanting province approved the study before launching the survey: Comité de Ética de la Investigación de la Provincia de Jujuy (Approval ID Not aplicable); Comisión Provincial de Investigaciones Biomédicas de la Provincia de Salta (Approval ID 321-284616/2019); Consejo Provincial de Bioética de la Provincia de La Pampa (Approval ID Not aplicable); Comité de Ética Central de la Provincia de Buenos Aires (Approval ID 2919-2056-2019). Informed consent was obtained from all the participants before answering the survey. All aspects of the study were conveyed to the participants prior to collecting consent, including details of anonymity and the confidentiality of the data collected. Special emphasis was placed on the precautions taken to secure and de-identify data, the respondents' ability to withdraw at any time, and data protection procedures.

## Findings

In total, 87 providers responded to the survey. The consent rate defined as the number who consented /the number of eligible providers (89/112) was 79.5%. The main reasons why some providers declined to participate were linked to being overburdened to provide health care response to COVID-19, or, being themselves infected with COVID-19. Among people who consented, almost all of them completed the survey (87/89, 97.8%). Key provider characteristics are presented in Table 1. Most respondents (41.4% of the sample) were from province 2, and most provided services at the tertiary level (66.7%). Most respondents (57.5%) were between 30 to 44 years old, and female (65.5%). In addition, 40.2% of respondents had between 10 to 20 years of experience.

Almost all respondents correctly identified which grounds were currently legal for abortion in Argentina. Proportions varied depending on the ground: for saving the woman's life, 96.6% responded correctly; for preserving the woman's health, 89.7% did so; and in case of rape, 87.4%. Finally, 81.6% of respondents knew that abortion on request is currently legal (10.3% of participants refused to respond). Despite indicating that these grounds were legal in Argentina, 39.3% of respondents were unwilling to perform an abortion to save the woman's life; 43.6% refused to perform an abortion to preserve the woman's health; 46.1% would not perform an abortion in cases of rape; and 53.5% responded that they were unwilling to perform an abortion based on a woman's request. Most providers responded that their reason for being unwilling to perform the practice was based on religious, or moral reasons (i.e., conscientious objection). Proportions of respondents who gave this reason ranged from 78.8%-91.2%, depending on the legal ground. Among those unwilling to perform an abortion who did not cite conscientious objection, other reasons were cited and the explained as "practicing another specialty in the facility", and "not receiving requests for abortions" (Table 2).

Then, we compared responses from willing and unwilling providers to explore any differences in perceptions of additional restrictions or requirements under the new law between them (Table 3). Most respondents knew that there was a gestational age limit for abortion on request: 93.3% of those willing and 81.6% of those unwilling to perform abortions identified

**Table 1. Provider characteristics.**

|  | (N = 87) n | % |
| --- | --- | --- |
| **District** | | |
| 1 | 26 | 29.9 |
| 2 | 36 | 41.4 |
| 3 | 9 | 10.3 |
| 4 | 16 | 18.4 |
| **Facility type: Primary Care** | | |
| Yes | 17 | 19.5 |
| No | 70 | 80.5 |
| **Facility type: Secondary Care** | | |
| Yes | 17 | 19.5 |
| No | 70 | 80.5 |
| **Facility type: Tertiary Care** | | |
| Yes | 58 | 66.7 |
| No | 29 | 33.3 |
| **Age (years)** | | |
| <30 | 6 | 6.9 |
| > = 30 and <45 | 50 | 57.5 |
| > = 45 and < = 60 | 24 | 27.6 |
| Refused | 7 | 8.0 |
| **Gender** | | |
| Male | 28 | 32.2 |
| Female | 57 | 65.5 |
| Refused | 2 | 2.3 |
| **Number of years in practice** | | |
| <10 | 30 | 34.5 |
| > = 10 and <20 | 35 | 40.2 |
| > = 20 and < = 42 | 18 | 20.7 |
| Refused | 4 | 4.6 |

this restriction, but no statistically significant differences were found between groups. In contrast, gestational age limits have not been stipulated by law for the rest of the grounds but some providers in both groups believed they had.

Similarly, judicial authorization for minors (not required by law) was erroneously perceived to be a requirement by 7.7%-20.5% of willing providers and 22.9%-35.3% of unwilling ones. Differences in this perception between both groups were statistically significant in the case of preserving a woman´s health (p = 0.019) and in the case of rape (p = 0.016).

Also, compulsory counseling (defined as a counseling provided to pregnant people requesting an abortion whose sole purpose is to dissuade pregnant people from having an abortion) was believed to be a requirement by 13.3%-18.6% of willing providers and by 34.4%-47.2% of unwilling professionals across the four grounds. However, in the case of abortion on request, unwilling providers were more likely to report this additional requirement, and the difference between groups of providers was statistically significant (p = 0.007). The requirement of an HIV test was perceived to be a requirement by 7.7%-17.1% of willing providers and by 22.6%-45.5% of unwilling ones. For this additional requirement, statistically significant differences were found in the case of rape (p = 0.024) and for abortions on request (p = 0.005) between both groups of professionals.

**Table 2. Providers´ knowledge of current legal grounds currently for abortion under national law, willingness to provide induced abortion, and reasons for non-performance.**

| | To save a woman's life | | To preserve a woman's health | | In cases of rape | | On request | |
|---|---|---|---|---|---|---|---|---|
| | n | % | n | % | n | % | n | % |
| **Ground is legal?** | | | | | | | | |
| No | 1 | 1.1 | 0 | 0.0 | 1 | 1.1 | 4 | 4.6 |
| Yes | 84 | 96.6 | 78 | 89.7 | 76 | 87.4 | 71 | 81.6 |
| Don't know | 1 | 1.1 | 1 | 1.1 | 3 | 3.4 | 3 | 3.4 |
| Refused | 1 | 1.1 | 6 | 6.9 | 7 | 8.0 | 9 | 10.3 |
| Missing | 0 | 0.0 | 2 | 2.3 | 0 | 0.0 | 0 | 0.0 |
| **Willingness to provide abortion care** | | | | | | | | |
| No | 33 | 39.3 | 34 | 43.6 | 35 | 46.1 | 38 | 53.5 |
| Yes | 44 | 52.4 | 26 | 33.3 | 36 | 47.4 | 30 | 42.3 |
| Don't know | 0 | 0.0 | 1 | 1.3 | 3 | 3.9 | 2 | 2.8 |
| Refused | 5 | 6.0 | 0 | 0.0 | 2 | 2.6 | 1 | 1.4 |
| Missing | 2 | 2.4 | 17 | 21.8 | 0 | 0.0 | 0 | 0.0 |
| **If No, why not?** | | | | | | | | |
| Personal or religious or moral reason | 26 | 78.8 | 31 | 91.2 | 30 | 85.7 | 32 | 84.2 |
| Facility reason: religious | 0 | 0.0 | 0 | 0.0 | 0 | 0.0 | 1 | 2.6 |
| Facility reason: clinical capacity | 1 | 3.0 | 0 | 0.0 | 1 | 2.9 | 1 | 2.6 |
| Other | 4 | 12.1 | 2 | 5.9 | 1 | 2.9 | 2 | 5.3 |
| Refused | 1 | 3.0 | 0 | 0.0 | 1 | 2.9 | 1 | 2.6 |

**Table 3. Providers´ perceptions of additional requirements-restrictions between willing and unwilling providers.**

| | Willingness to provide abortion care | | | | | | | | | | | |
|---|---|---|---|---|---|---|---|---|---|---|---|---|
| **Legal ground** | To save a woman's life | | | To preserve a woman's health | | | In cases of rape | | | On request | | |
| **Additional requirements-restrictions** | Yes (n = 44) % | No (n = 33) % | yp* | Yes (n = 26) % | No (n = 34) % | p* | Yes (n = 36) % | No (n = 35) % | p* | Yes (n = 30) % | No (n = 38) % | p* |
| Gestational limits *(only applies for abortion on request)* | 22.7 | 30.3 | 0.626 | 42.3 | 32.4 | 0.601 | 36.1 | 34.3 | 1.000 | 93.3 | 81.6 | 0.468** |
| **Additional requirements-restrictions not stipulated by law** | | | | | | | | | | | | |
| Authorization of health care professional required | 27.3 | 51.5 | 0.053 | 34.6 | 52.9 | 0.249 | 22.2 | 45.7 | 0.066 | 20.0 | 43.2 | 0.080 |
| Judicial authorization for minors | 20.5 | 30.3 | 0.469 | 7.7 | 23.5 | 0.019** | 8.6 | 35.3 | 0.016 | 13.3 | 22.9 | 0.505 |
| Authorized in specially licensed facilities only | 36.4 | 51.5 | 0.273 | 34.6 | 61.8 | 0.068 | 55.6 | 64.7 | 0.591 | 50.0 | 54.5 | 0.914 |
| Compulsory counseling | 18.6 | 34.4 | 0.199 | 15.4 | 42.4 | 0.051 | 14.7 | 35.3 | 0.093 | 13.3 | 47.2 | 0.007 |
| Compulsory waiting period | 11.4 | 9.4 | 0.187** | 11.5 | 20.6 | 0.143** | 11.1 | 20.6 | 0.447 | 13.3 | 25.0 | 0.381 |
| Parental consent required for minors | 45.5 | 54.5 | 0.576 | 57.7 | 52.9 | 0.917 | 48.6 | 41.2 | 0.707 | 56.7 | 51.4 | 0.851 |
| HIV Tests | 16.7 | 22.6 | 0.739 | 7.7 | 29.4 | 0.079 | 17.1 | 45.5 | 0.024 | 10.0 | 44.4 | 0.005 |

*Proportional z-test

**Fisher's exact test

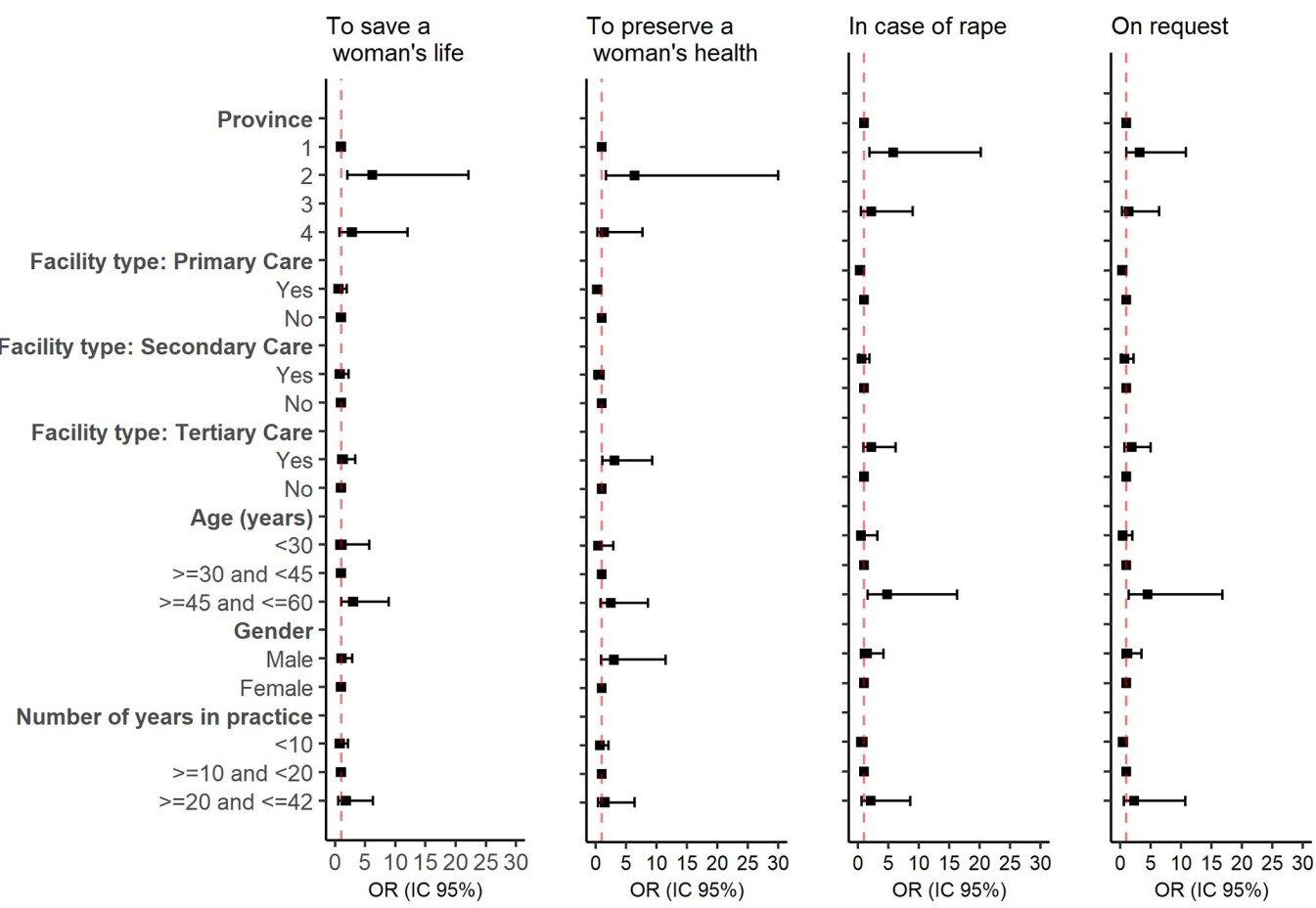

**Fig 1. Odds ratio of the association between provider´s socio-demographic characteristics and the unwillingness to performing induced abortions.**
Dotted red line shows an OR equal to 1.

Fig 1 presents the results of the analysis to evaluate which provider socio-demographics characteristics were associated with unwillingness to provide abortion care.

For the ground "to save a woman's life", the province where the provider works was the only factor associated with refusal to provide care (province 2 compared to province 1 increased the odds by 6.2 times: (OR 6.2 [CI 95%: 2.0–22.1] (p = 0.006)). For the ground "to preserve the woman's health", the province and the facility type in which the provider was working were both significantly associated with unwillingness to perform abortions. Working in province 2 compared to province 1 increased the odds of unwillingness to perform abortions by 6.4 times: (OR 6.4 [CI 95%: 1.7–30.0] (p = 0.018)), while working in the primary care decreased the odds of unwillingness to perform abortions by 0.2 times (OR 0.2 [CI 95%: 0.1–0.9] (p = 0.033)) and in a tertiary-level facility increased the odds by 3.1 times (OR 3.1 [CI 95%: 1.1–9.3] (p = 0.033)). The province in which the provider worked and the fact or not of working in the tertiary level remained significant in the multivariate model.

For the legal ground "in case of rape", the province and the age of the provider were significantly associated with unwillingness to provide an abortion. Working in province 2 increased the odds of being unwilling to perform abortions by an OR of 5.8 [CI 95%: 1.9–20.2] (p = 0.009) and being 45 years old or more by an OR of 4.8 [CI 95%: 1.6–16.3] (p = 0.008) compared with those aged between 30–45 years old. When including the two variables in the

multivariate model, the age of the provider was no longer significant. In the case of the ground "on request", the age of the providers and the number of years in practice were associated with unwillingness to provide care. Being 45–60 years old increased the odds of unwillingness to provide an abortion by 4.5 times compared with being 30–45 years old, with an OR of 4.5 [CI 95%: 1.4–16.8] p = 0.010. Having more than 20 years of practice raised the odds of unwillingness to provide an abortion into 6.4 times compared with those with less than 10 years [CI 95%: 11.6–31.1] p = 0.022. In the multivariate analysis, the age of provider was the only variable that remained significance due to the presence of collinearity between the two variables (S1–S5 Tables).

## Discussion

Our study aimed to assess knowledge of the current legal framework and willingness to perform abortions among authorized professionals authorized to do it in Argentina, to compare perceptions about requirements for performing abortion that are stipulated by law between willing and unwilling providers, and to explore whether there are demographic factors associated with unwillingness to provide abortion services in the context of the recent removal of legal restrictions. Our results show that most providers correctly identified the grounds upon which abortions are currently legal in Argentina, but despite legality, around half of respondents stated they were unwilling to perform abortions, mainly due to conscientious objection. Both willing and unwilling providers endorsed additional requirements that were not legally stipulated, although these misinterpretations were higher in the unwilling group. Province, type of facility, and the age of the provider were associated with unwillingness to provide abortion care on certain grounds. Our results indicate that barriers to abortion provision at provider level may constitute an important obstacle to abortion access in the context of the new favorable legal framework in Argentina.

There is evidence that supports the existence of knowledge gaps of abortion laws and related policies by health providers and even by policy makers, globally [15]. In Argentina, this problem was detected years ago by a research study conducted in public hospitals where the majority of professionals authorized to provide abortions could not recognize all of the legal grounds allowed at that time, showing a weak knowledge of the legal framework [17]. Surprisingly, in our study, we found that most health care providers could identify the grounds allowed under the new law. This change in providers´ knowledge related to legislation may be due to strong recent exposure to the debate on the decriminalization of abortion in society and therefore within health institutions. Undoubtedly, this intense exposure in the media and in society is a result of the impact and influence of feminist movements, not only in Argentina but also in the Region, expanding a truly Latin American green wave which after many years of activism and massive mobilizations, managed to put the issue of abortion decriminalization on the political and social agenda. At a global level, evidence on provider knowledge of legal grounds for abortion in their settings has yielded uneven results. For example, while in Ghana practitioners recognized the grounds allowed under the law [29], in Mexico and Nepal health care providers did not have sufficient knowledge about which cases were legally recognized in their contexts [12, 13]. In any case, professionals authorized to perform abortions may know when an abortion is legal but may not be fully aware of the presence (or absence) of requirements stipulated in the laws and associated policies.

Our research showed that although professionals could identify the grounds allowed under the new law, most respondents had misperceptions about requirements that are not currently legally stipulated. In a subgroup analysis, we found that proportions of incorrect answers were higher among those who declared themselves unwilling to perform abortions. For certain legal

grounds we detected statistically significant differences for some of these perceptions, such as judicial authorization for minors, requirements that abortions be performed in facilities with special license/authorization, compulsory counselling, and requiring HIV testing as a pre-condition for obtaining an abortion. This finding could reflect a poor generalized understanding of the law, lack of exposure to a new national clinical guideline or standard operating protocol [9] that explains the procedures to be followed in performing abortions under the new law more than a relevant difference in knowledge between both groups. These findings are consistent with the World Health Organization's (WHO) observations that the interpretations of legislations based on gestational age and grounds may lead problems related to differences in interpretation among providers [30]. These differences in interpretation can lead to errors in ascertaining the eligibility of individuals seeking abortions [31] and also in difficulties for operationalizing the implementation of WHO's new global guidelines.

It may be that some of the misinterpretations observed might be related to actions that providers perceived to be part of good clinical practice, rather than as mandatory legal requirements that are preconditions for abortion, such as requesting an HIV test before performing an abortion in the case of rape. Other responses might reflect confusion about the terminology used, as the case of "compulsory counseling" versus "options counseling". The former is not mentioned in the national guideline, whereas the latter is indicated as part of the protocol for abortion care. These two terms have totally different meanings, with the former is intended to dissuade a woman from having an abortion and the latter seeking to provide the woman with uncoercive health information. Likewise, a recent study conducted in Ireland, a country that similarly to Argentina has recently legalized abortion, showed that just 25% of the participants correctly identified all the requirements stipulated by the new law [11]. In other settings, health professionals also report confusion and uncertainty about abortion-related laws and policies; there is particular confusion about the documentation required to provide abortion care in cases of rape, and the requirement for authorization by a consultant heath care provider [12, 13].

These findings can inform potential actions that may improve the implementation of the law and associated guidelines, not only in Argentina, but globally. Healthcare providers need to be educated on the legal grounds for abortion as well as associated requirements. Implementation strategies should be developed that include provider support through training, continuing medical education, values clarification interventions [32], and implementation tools (e.g., checklists, screening tools) that help the provider to correctly apply requirements or restrictions, especially in contexts where there has been recent legalization and/or where there are numerous requirements and misinterpretations of the law or omissions could lead to adverse results.

In our study, between 39% and 54% of the respondents answered that they were unwilling to perform abortions mainly for being conscientious objectors. Our results are similar to those of a previous survey conducted in Argentina prior to the approval of abortion on request, in which 50% of participants, and even more for some legal grounds, declared themselves unwilling to perform abortions [17]. Thus, our results are consistent with background literature demonstrating that conscientious objection represents an important barrier to access to legal abortion, one that poses a threat to implementation of the new law in Argentina. Other countries in the region report similar problems. In Uruguay, where abortion was decriminalized in 2012, about 30% of Uruguayan obstetricians and gynecologists have declared conscientious objector status [33].

However, misuses of conscientious objection may account for at least some of the cases in Argentina and globally [18, 20, 34]. Misuses include instances of providers refusing to provide care for other reasons, while claiming moral or ethical beliefs to avoid performing abortions. Such refusal may be motivated by ignorance about the domestic laws and policies that regulate

the practice, fear of potential legal problems, stigma from colleagues, and the influence of facility leadership affecting professionals' ability to make independent decisions about performing abortions on legal grounds, among others [13, 20, 35]. When a large proportion of professionals invoke conscientious objection, ensuring the existence of mechanisms for timely and proper referral to other authorized professionals or ensuring that facilities have enough willing providers is critical. In this regard, WHO has issued recommendations encouraging states to issue regulations and policies designed to increase the spectrum of health professionals (i.e., general practitioners, midwives, nurses) with skills and authorization to safely provide abortions, and to ensure that sufficient willing providers are employed and equitably distributed within the health system [30].

In our study, some providers' demographic factors, such as the province where they work, employment at a tertiary-level facility, and older age, were associated with unwillingness to perform abortions on some grounds. Consistent with our results, a survey that analyzed predictors of conscientious objection and abortion willingness across different clinical scenarios, showed that unwillingness to provide service varies by clinical scenarios and by some physician characteristics such as region where they worked or being older [36]. Possible reasons for these associations could include, for example, that some provinces where providers receive their medical training and socialization offer more restrictive environments. With respect to age, younger professionals might be more willing to perform abortions because they are more likely to have been exposed to social and cultural changes aimed at improving access to the practice.

## Limitations

The main limitations of this study are linked to the small sample size. This issue is reflected in the wide confidence intervals around our results. Due to the small sample size, our findings should be considered exploratory; the analysis may lack sufficient power to detect significant differences between willing and unwilling providers. Further research with a larger sample is needed to make more robust conclusions and, if confirmed, to drive interventions.

Consent rates to participate were generally high (90% in province 1, 78% in province 2, and 89% in province 4). However, in province 3, we obtained a low consent rate (64%) which could have generated selection bias, if those who did not consent to participate were more likely to be unwilling providers. Providers may have been particularly unwilling to participate in the survey because of its temporal proximity to the change in the law; the multi-year campaign leading up to the change was heavily politized. Moreover, this study was conducted during the height of the COVID pandemic when providers were fully focused on the pandemic response, which contributed to delays in administering the surveys and obtaining responses. We implemented several strategies to increase recruitment, such as administering the survey on paper and in electronic format and extending the time stipulated for the study. Additionally, the general sensitivity of the topic may also have contributed to low participation, even though we clearly explained the measures taken to maintain the confidentiality of the responses. Also, our study sample underrepresents respondents from primary care facilities, who according to our results are more likely to be willing to do the practice than those working in tertiary care facilities. Another limitation of our study is that we have not asked about training in abortion care under the scope of the new law. This would have allowed us to identify potential gaps in interpretations and even in care skills.

## Strengths

Our study has several notable strengths. First, this survey was conducted immediately after passage of the new law, and as such our results can be useful as a baseline for future

comparisons, understanding, that the short exposure time may have had an impact on the degree of awareness of the changes introduced. Additionally, the multi-stage sampling methodology, in four provinces of the country, gives diversity to the sample. The selection criteria for participants included all providers legally authorized to perform abortions rather than solely those who currently provide the service, and thus allowed us to compare responses collected from those who were willing as well as those unwilling to provide abortions and to identify potential provider-level barriers to abortion access among both groups. Finally, the self-administration approach and the confidential nature of the survey may have averted interviewer bias.

## Conclusions

Through a confidential self-administered survey, we showed that most providers authorized to perform abortions know when an abortion is legal, yet almost half of them would be unwilling to perform the practice due to conscientious objection. Both willing and unwilling providers were not fully aware of the requirements stipulated in the law and associated protocols, although misinterpretations were higher in the unwilling group. Finally, we found that the province where a provider serves, employment in tertiary-level facilities, and older age were associated factors to unwillingness for certain legal grounds. The results of our study indicate that, even in the new favorable legal context, barriers at the provider level may hinder access to abortion in Argentina. It will be important to design provider-level strategies that include broad dissemination of the correct interpretation of the new law and protocols for its implementation, and to design tools that can support providers, as well as clarify who may object and under what circumstances. Moreover, public policies should be aimed at the organization of health services to ensure an effective number of willing providers so that abortion seekers can access abortion services that are currently permitted by law.

## Supporting information

**S1 Checklist. STROBE statement—checklist of items that should be included in reports of *cross-sectional studies*.**
(DOC)

**S1 Table. Associated factors with unwillingness to performing induced abortions.**
(DOCX)

**S2 Table. Associated factors with unwillingness to performing induced abortions to save a woman's life.**
(DOCX)

**S3 Table. Associated factors with unwillingness to performing induced abortions to preserve a woman's health.**
(DOCX)

**S4 Table. Associated factors with unwillingness to performing induced abortions in case of rape.**
(DOCX)

**S5 Table. Associated factors with unwillingness to performing induced abortions on request.**
(DOCX)

**S1 File. Inclusivity in global research.**
(DOCX)

## Acknowledgments

The authors would like to thank the following people, without whose efforts the publication of this manuscript would not have been possible. We gratefully acknowledge the compromise and dedication of the provincial teams, members of the Maternal and Child Health Programs of the Provincial Ministries of Health: Dr. Daniel Nowacky, Dr. Adriana Allones, Marta Ferrary, Ana Seimande, Antonio Tabarcachi, Noelia Coria, Laura Soto, Dr. Mara Bazán, Dr. Patricia Leal, and Marcela Tapia. We also want to thank Dr. Gabriela Perrotta for her valuable consulting on abortion, and to Alvaro Ciganda and Julieta Spagnuolo for their guidance in data management. Finally, we would like to express our deepest gratitude to all of the health workers who participated in the study as data collectors, working through the height of the COVID-19 pandemic in Argentina.

## Author Contributions

**Conceptualization:** Paula Vázquez, Carolina Nigri, Verónica Pingray, Caitlin R. Williams, Mabel Berrueta.

**Data curation:** Paula Vázquez, Carolina Nigri, Verónica Pingray, Luz Gibbons, Sandra Formia, Jewel Gausman, Caitlin R. Williams, Mabel Berrueta.

**Formal analysis:** Luz Gibbons.

**Funding acquisition:** Ana Langer, R. Rima Jolivet.

**Investigation:** Paula Vázquez, Analía Messina, Claudia Castro, Cintia Jacobi, Adriana Martiarena, Susana Velazco, Caitlin R. Williams.

**Methodology:** Paula Vázquez, Carolina Nigri, Verónica Pingray, Luz Gibbons, Caitlin R. Williams, Mabel Berrueta.

**Project administration:** R. Rima Jolivet.

**Supervision:** Mabel Berrueta.

**Writing – original draft:** Paula Vázquez.

**Writing – review & editing:** Paula Vázquez, Carolina Nigri, Verónica Pingray, Sandra Formia, Analía Messina, Claudia Castro, Cintia Jacobi, Adriana Martiarena, Susana Velazco, Ana Langer, Jewel Gausman, R. Rima Jolivet, Caitlin R. Williams, Mabel Berrueta.

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
