## [Decision Letter · Decision Letter 0]

22 May 2023

PONE-D-23-07008"Factors associated with provider unwillingness to perform induced abortion in Argentina: A cross-sectional study in four provinces following the legalization of abortion on request"PLOS ONE

Dear Dr. Vázquez,

Thank you for submitting your manuscript to PLOS ONE. After careful consideration, we feel that it has merit but does not fully meet PLOS ONE’s publication criteria as it currently stands. Therefore, we invite you to submit a revised version of the manuscript that addresses the points raised during the review process. The article is scientifically valid and well structured; only minor revisions are needed as indicated by reviewers.

We look forward to receiving your revised manuscript.

Kind regards,

Andrea Cioffi

Academic Editor

PLOS ONE

Reviewers' comments:

Reviewer's Responses to Questions

**Comments to the Author**

1. Is the manuscript technically sound, and do the data support the conclusions?

Reviewer #1: Yes

Reviewer #2: Yes

Reviewer #3: Yes

2. Has the statistical analysis been performed appropriately and rigorously? 

Reviewer #1: Yes

Reviewer #2: Yes

Reviewer #3: Yes

3. Have the authors made all data underlying the findings in their manuscript fully available?

Reviewer #1: Yes

Reviewer #2: Yes

Reviewer #3: Yes

4. Is the manuscript presented in an intelligible fashion and written in standard English?

Reviewer #1: Yes

Reviewer #2: Yes

Reviewer #3: Yes

5. Review Comments to the Author

Reviewer #1: The authors showed that most providers authorized to perform abortions know when an abortion is legal, yet almost half of them would be unwilling to perform the practice due to conscientious objection. Their results indicate that barriers at the provider level may hinder access to abortion in Argentina. According to the results the authors suggest broad dissemination of the correct interpretation of the new law and protocols for its implementation, to design tools that can support providers, and to ensure an effective number of willing providers so that abortion seekers can access abortion services. This is an interesting topic and worth of further exploring.

Reviewer #2: General

The paper addresses an essential knowledge and implementation gap regarding the refusal to provide care by authorized health providers after the change of abortion law in Argentina in 2020. This paper adds to an increasing body of literature exploring the motivations to declare conscientious objection when asked to provide abortion care, among other potential access barriers for people seeking abortion care. The authors use appropriate citations and methods to address the research question. As potential areas of improvement, it may be helpful to include more citations and examples of studies done in other parts of Latin America and the world addressing this issue, especially considering the recent law change. Please see the below comments for more detail.

General comments

We suggest the following during the text:

• We suggest revisiting the Settings section, as the explanation is confusing.

• We recognize that you mentioned that provinces' names were removed for confidentiality reasons. However, we suggest you consider including provinces' characteristics, so the reader can better understand the relationship between unwillingness and the context of providers.

• Clarify if the concept of conscientious objection was defined to the participants in the survey or the consent form. As it is a concept, as you mentioned, that can be used in different ways, it is essential to recognize the potential interpretations as a limitation in case it is not defined.

• Clarify if health professionals had any training in abortion care or if you asked for that information, as it can also be a limitation of your study.

Specific suggestions

• Line 63 (Page 3): We suggest presenting Methods and Findings/Results separately.

• Line 72 (Page 3): Instead of using “They suggest that to assure abortion rights (…)” We suggest you moderate the conclusions by showing that those results can help inform specific actions (training and future research, for example) that can improve abortion access.

• Line 99 (Page 5): Is there any other example or international experience that can support the sentence “The law is expected to improve public health, as laws legalizing abortion create enabling environments for skilled providers and the access to safe methods”.

• Line 102 (Page 5): As the world abortion environment has changed dramatically since 2021 (legalization in Colombia and restrictions in the US, for example) we suggest you update this affirmation.

• Line 108 (Page 5 and in the entire paper): We suggest you use either “Pregnant people” or “women” (the latter specifying that includes trans and non-binary people) instead of using both throughout the paper.

• Line 118 (Page 6): Is there any other study that can support this sentence?

• Line 121 (Page 6): Health personnel authorized to provide abortions may vary according to the legal stipulations and health system. It may be necessary to briefly mention who is included in this “provider authorized to perform an abortion” group in Argentina.

• Line 124 (Page 6): Legal requirements to perform abortion may vary in each country and region. It may be necessary to include which are the ones that apply in Argentina, to help the reader understand the findings and conclusions.

• Line 194 (Page 9): As mentioned in the section above, when you say “abortion practices” are you referring to medical abortion, surgical abortion, or both? Please clarify here and on the entire paper.

• Line 198 (Page 9): Please specify if there are other exclusion criteria.

• Line 283 (12): When addressing the years of experience, do you also have information about the specific practice they have experience in? Would you consider that a relevant element for your analysis and their willingness to provide abortion? (For example, General practitioner vs. OBYGN)

• Table 2 (Page 15): How are “Personal or religious or moral reason” and “Facility reason: religious” different? Considering that you mentioned that in Argentina the conscientious objection couldn’t be institutional.

• Line 335 (Page 18): Figure 1 is not in the body of the paper.

• Line 345 (Page 18): When hypothesizing about the relationship between the province and willingness to provide service, don’t you consider it relevant to include the province’s characteristics that may explain better those potential connections? This could also help create specific recommendations for future actions.

• Line 425 (Page 21): We suggest editing the sentence to show that these findings can inform potential actions instead of assuming the results suggest actions just as they are.

• Line 485 (Page 23): Can this – the survey being conducted immediately after passage of the new law – also be a limitation of your study?

Reviewer #3: Thank for you the opportunity to review "Factors associated with provider unwillingness to perform induced abortion in Argentina: A cross-sectional study in four provinces following the legalization of abortion on request." This is a very useful study and valuable contribution to the literature. Overall the article is well structured and results fairly well situated within Argentina’s broader socio-legal context leading to the change in legal status. A few recommendations are suggested prior to publication.

Introduction

Relevance of context and social movements for legalization is critical, however not well explained. An additional sentence or two pertaining to grassroots mobilization for “social and cultural change” would benefit readers who are unfamiliar with the national context (line 98-99).

Line 120 – I appreciate the study is limited to legally authorized providers, but other health system actors’ knowledge of abortion laws and policies have implications. Data or interrogation about ways other health system actors can enable/constrain access to abortion services within the Argentinian context would be beneficial here; especially since equitable labour distribution is expanded on and recommended in the discussion.

Data collection

Data collection protocols are clear and detailed. Confirming that recruitment only involved a one-time email sent to potential participants? Any follow-up efforts for non-respondents?

Findings

The findings are presented effectively and well-support by the accessible tables and figures.

Participants refusal to respond to whether abortion on request is a legal ground (10.3%) and willingness to provide abortion to save a woman’s life (6%) are interesting findings. Curious whether non-response to the latter questions is also associated with province, facility level and age variables in any way? If authors have analysis or inferences to draw from this (or situate it within the broader literature) it would be interesting to unpack in the discussion.

Discussion

Line 386 – Generally, but also specifically because authors highlight how the results differ from prior work on provider legal knowledge, the contemporary socio-legal context of Argentina needs to be fleshed out further. The influence and impact of the Latin American Green Wave and Causa Justa movement cannot be understated at both regional and country level. Expounding here on its relevance to the findings may better support the author’s comparison between the 2014 and 2023 studies.

Line 406 – Regarding analysis on clinical guidelines and SOPS, any implications here for steps to operationalize WHO’s updated Abortion Care Guidance, which recommends against grounds-based approaches in law and policy?

Given the focus on CO in the findings, I wondered if authors considered implementation strategies beyond checklist/widget approaches to improve accurate application of the law (line 428)? Recommendations for (continuing) medical education, curricula development, values clarification and attitudes transformation modalities, etc. may be particularly relevant here.

Line 469 – adding a sub-heading for strengths/limitations may be useful for flow.

Thank you for the opportunity to review this manuscript. I believe the PLOS ONE editors are well positioned to ensure the above comments are addressed. Please note I believe in a transparent review process and have provided the authors and the journal editors with the same comments.

6. PLOS authors have the option to publish the peer review history of their article (what does this mean?). If published, this will include your full peer review and any attached files.

Reviewer #1: No

Reviewer #2: **Yes: **Paula Pinzon MPH, Supervised by Professor Wendy Norman, UBC, Canada

Reviewer #3: No

---

## [Author Response · Author response to Decision Letter 0]

5 Aug 2023

PONE-D-23-07008

"Factors associated with provider unwillingness to perform induced abortion in Argentina: A cross-sectional study in four provinces following the legalization of abortion on request"

PLOS ONE

Dear Andrea Cioffi, Academic Editor, PLOS ONE:

Please find in this document our responses to the review feedback received. This rebuttal letter responds to each point raised by the academic editor and reviewer(s). We will upload this letter as a separate file labeled 'Response to Reviewers'.

We hope that our responses will meet your expectations and those of the other reviewers. Please let us know if further corrections are needed to strengthen the manuscript.

We also uploaded the following:

● A marked-up copy of our manuscript that highlights changes made to the original version. We uploaded this as a separate file labeled 'Revised Manuscript with Track Changes'.

● An unmarked version of our revised paper without tracked changes. We uploaded this as a separate file labeled 'Manuscript'.

● A PLOS’ questionnaire on inclusivity in global research (as a Supporting Information file) 

● Figure 1 (Fig1.tiff) with Preflight Analysis and Conversion Engine (PACE) digital diagnostic tool (tested)

On behalf of all co-authors, best regards,

Paula Vázquez

 

RESPONSE TO REVIEWERS 

Note: (Responses are in green. New information added in red. Lines numbers are referred to Revised Manuscript with Track Changes).

Response to the Academic Editor:

Q: The article is scientifically valid and well structured; only minor revisions are needed as indicated by reviewers.

A: Thank you for your positive appraisal. We did our best to improve the manuscript after receiving the constructive comments from you and the reviewers. 

Q: A: There are no changes necessary to the financial disclosure statement. All authors declare no competing interests exist.

Q: Guidelines for resubmitting your figure files are available below the reviewer comments at the end of this letter.

A: Thank you for this guidance. We have reviewed and followed the guidelines for resubmission. Furthermore, we uploaded edited figure (Fig 1) with the recommended Preflight Analysis and Conversion Engine (PACE) tool. Let us know if any requirement has to be revisited again. 

Q: If applicable, we recommend that you deposit your laboratory protocols in protocols.io to enhance the reproducibility of your results. Protocols.io assigns your protocol its own identifier (DOI) so that it can be cited independently in the future. For instructions see: https://journals.plos.org/plosone/s/submission-guidelines#loc-laboratory-protocols. Additionally, PLOS ONE offers an option for publishing peer-reviewed Lab Protocol articles, which describe protocols hosted on protocols.io. Read more information on sharing protocols at https://plos.org/protocols?utm_medium=editorial-email&utm_source=authorletters&utm_campaign=protocols.

A: Thank you for this information. All our data have been anonymized to ensure compliance with human subject protections and study protocols. The anonymized data underlying the findings are deposited in the Harvard Dataverse repository with a unique doi number and URL, which we have provided.

Q1. Please ensure that your manuscript meets PLOS ONE's style requirements, including those for file naming. The PLOS ONE style templates can be found at

A: We have reviewed PLOS ONE’s style requirements and ensured that the manuscript complies.

Q2. Please include a complete copy of PLOS’ questionnaire on inclusivity in global research in your revised manuscript. Our policy for research in this area aims to improve transparency in the reporting of research performed outside of researchers’ own country or community. The policy applies to researchers who have travelled to a different country to conduct research, research with Indigenous populations or their lands, and research on cultural artefacts. The questionnaire can also be requested at the journal’s discretion for any other submissions, even if these conditions are not met. Please find more information on the policy and a link to download a blank copy of the questionnaire here: https://journals.plos.org/plosone/s/best-practices-in-research-reporting. Please upload a completed version of your questionnaire as Supporting Information when you resubmit your manuscript.

A: We have completed the requested questionnaire and uploaded as supporting information. 

Q3. We note that you have included the phrase “data not shown” in your manuscript. Unfortunately, this does not meet our data sharing requirements. PLOS does not permit references to inaccessible data. We require that authors provide all relevant data within the paper, Supporting Information files, or in an acceptable, public repository. Please add a citation to support this phrase or upload the data that corresponds with these findings to a stable repository (such as Figshare or Dryad) and provide and URLs, DOIs, or accession numbers that may be used to access these data. Or, if the data are not a core part of the research being presented in your study, we ask that you remove the phrase that refers to these data.

A: We clarify consent rates in the four provinces as follows: 

Lines 595-596: 

Additionally, we obtained a low consent rate in province 3 (64%, data not shown), Consent rates to participate were generally high (90% in province 1, 78% in province 2, and 89% in province 4). However, in province 3, we obtained a low consent rate (64%) which could have generated selection bias, if those who did not consent to participate were more likely to be unwilling providers.

Q4. Please review your reference list to ensure that it is complete and correct. If you have cited papers that have been retracted, please include the rationale for doing so in the manuscript text, or remove these references and replace them with relevant current references. Any changes to the reference list should be mentioned in the rebuttal letter that accompanies your revised manuscript. If you need to cite a retracted article, indicate the article’s retracted status in the References list and also include a citation and full reference for the retraction notice.

A: We have done so in the specific responses appearing below.

Reviewers' comments:

Reviewer's Responses to Questions

Comments to the Author

1. Is the manuscript technically sound, and do the data support the conclusions?

Reviewer #1: Yes

Reviewer #2: Yes

Reviewer #3: Yes

2. Has the statistical analysis been performed appropriately and rigorously?

Reviewer #1: Yes

Reviewer #2: Yes

Reviewer #3: Yes

3. Have the authors made all data underlying the findings in their manuscript fully available?

Reviewer #1: Yes

Reviewer #2: Yes

Reviewer #3: Yes

4. Is the manuscript presented in an intelligible fashion and written in standard English?

Reviewer #1: Yes

Reviewer #2: Yes

Reviewer #3: Yes

5. Review Comments to the Author

Reviewer #1: The authors showed that most providers authorized to perform abortions know when an abortion is legal, yet almost half of them would be unwilling to perform the practice due to conscientious objection. Their results indicate that barriers at the provider level may hinder access to abortion in Argentina. According to the results the authors suggest broad dissemination of the correct interpretation of the new law and protocols for its implementation, to design tools that can support providers, and to ensure an effective number of willing providers so that abortion seekers can access abortion services. This is an interesting topic and worth of further exploring.

Thank you for your comment, we hope to conduct further research on this topic that will allow us to capture future changes in the attitudes of professionals regarding abortion practices.

Reviewer #2: General

The paper addresses an essential knowledge and implementation gap regarding the refusal to provide care by authorized health providers after the change of abortion law in Argentina in 2020. This paper adds to an increasing body of literature exploring the motivations to declare conscientious objection when asked to provide abortion care, among other potential access barriers for people seeking abortion care. The authors use appropriate citations and methods to address the research question. As potential areas of improvement, it may be helpful to include more citations and examples of studies done in other parts of Latin America and the world addressing this issue, especially considering the recent law change. Please see the below comments for more detail. 

Thank you for your comments, we have responded in detail below. 

General comments

We suggest the following during the text:

• We suggest revisiting the Settings section, as the explanation is confusing.

A: We decided to simplify the text and refer the reader to the publication of the protocol of the master study where the authors describe in detail the characteristics of the sampling to select the provinces: 

Lines 220-223: 

Provinces/districts were selected according to a purposive sampling plan based on a composite index that was used as a proxy for health system performance in each study area, along with feasibility criteria reflecting the interest of the ministry/provincial government to participate in the study. The design of the index was described in the research protocol for the larger study [19] and took into account key maternal health indicators to reflect overall maternal health system performance. For Argentina, the index included maternal mortality ratio, number of prenatal visits, and the proportion of women receiving uterotonics at delivery. Based on this index, two rounds of selection were performed. In a first round, a region from the highest performing tercile of the index and one region from the lowest performing tercile were selected. In the second round, one highest-performing province and one lowest-performing province were selected from each region using the same index. Using this methodology, four provinces were selected. was described in the research protocol for the larger study [23] . Then, Finally, public facilities offering abortion services were selected from 20 randomly selected primary sampling units (PSU), following the same standardized multistage sampling plan used for the Demographic and Health Survey [20] [24] . Finally, as a result of this sampling, five facilities were selected in each province to represent the three levels of care: one tertiary care facility, one secondary care facility, and three primary care facilities. Given that the same sample of facilities selected served in the validation of other indicators of the master project, facilities not employing midwives were excluded. 

[23] Jolivet, R. R., Gausman, J., Adanu, R., Bandoh, D., Belizan, M., Berrueta, M., Chakraborty, S., Kenu, E., Khan, N., Odikro, M., Pingray, V., Ramesh, S., Saggurti, N., Vázquez, P., & Langer, A. (2022). Multisite, mixed methods study to validate 10 maternal health system and policy indicators in Argentina, Ghana and India: a research protocol. BMJ open, 12(1), e049685. https://doi.org/10.1136/bmjopen-2021-049685

[24] USAID. DHS Methodology [Internet]. The DHS Program. Demographic and Health Surveys. Methodology. [cited 2022 Jul 7]. Available from: https://dhsprogram.com/Methodology/Survey-Types/DHS-Methodology.cfm

• We recognize that you mentioned that provinces' names were removed for confidentiality reasons. However, we suggest you consider including provinces' characteristics, so the reader can better understand the relationship between unwillingness and the context of providers.

A: Thank you for pointing out this important issue. We have added a paragraph about provinces´ characteristics.

Lines 282-292: 

“However, we will mention some relevant indicators of the selected provinces (recorded for the year 2017, the latest data available at the time of initiating the selection of the provinces participating in the study) that would allow contextualizing the findings of this study. Of the four provinces selected, two belong to the Northwest region (province 2 and 4) and two to the Central region of Argentina (province 1 and 3). Maternal mortality ratio, was highest in provinces 2 and 4, (34 and 48 maternal deaths per 100,000 live births, respectively) almost double the values for province 1 and for province 3 (27 per 100,000 live births and 20 per 100,000 live births, respectively) and exceeding the national average of 29 per 100,000 live births [25]. In addition, the provinces representing the northwest had a total fertility rate that exceeded the national average value of 2.1, (ranging from 2.5 to 2.7), while in the provinces representing the Central Region the total fertility rate was lower than the national average [26]”.

[25] Ministerio de Salud y Desarrollo Social, Dirección Nacional de Sistemas de Información en Salud. Dirección de Estadísticas e Información en Salud. Estadísticas vitales. Información básica Argentina - Año 2017 [Internet]. Ministerio de Salud y Desarrollo Social. República Argentina; Available from: https://www.argentina.gob.ar/sites/default/files/serie5nro61.pdf

 [26] Instituto Nacional de Estadísticas y Censos (INDEC). Proyecciones provinciales de población por sexo y grupos de edad 2010-2040. Buenos Aires, Serie Análisis Demográﬁco N° 36 [Internet]. 2013. Available from: https://www.indec.gob.ar/ftp/cuadros/publicaciones/proyecciones_prov_2010_2040.pdf

• Clarify if the concept of conscientious objection was defined to the participants in the survey or the consent form. As it is a concept, as you mentioned, that can be used in different ways, it is essential to recognize the potential interpretations as a limitation in case it is not defined.

A: It is really a good point and we appreciate your input about this topic. The term conscientious objection was defined to the participants in the survey as “personal religious or moral reason” for which an authorized health provider may refuse to perform the practice. We added a sentence about this definition in the manuscript in the section Data collection and management. 

Lines 301-303: 

“In the survey, the term conscientious objection was defined to the participants as “personal religious or moral reason” for which professionals may opt out to perform the practice.”

• Clarify if health professionals had any training in abortion care or if you asked for that information, as it can also be a limitation of your study.

A: Thank you for raising this important issue, we have not asked about training in abortion care under the scope of the new law and we included this point as a limitation of our study. 

Lines 613-615: “Another limitation of our study is that we have not asked about training in abortion care under the scope of the new law. This would have allowed us to identify potential gaps in interpretations and even in care skills”. 

Specific suggestions

• Line 63 (Page 3): We suggest presenting Methods and Findings/Results separately.

A: Thank you for your suggestion. We have separated the two items (line 65 and line 68).

• Line 72 (Page 3): Instead of using “They suggest that to assure abortion rights (…)” We suggest you moderate the conclusions by showing that those results can help inform specific actions (training and future research, for example) that can improve abortion access.

A: Thank you for this constructive suggestion, we have revised the text and made changes that reflect that our results help to visualize the need for such specific actions.

Lines 75-78: “They help to demonstrate the need for specific actions that can improve access such as training, further research and public policies that guarantee facilities have sufficient professionals willing to provide abortion care”. 

• Line 99 (Page 5): Is there any other example or international experience that can support the sentence “The law is expected to improve public health, as laws legalizing abortion create enabling environments for skilled providers and the access to safe methods”. 

A: That sentence was supported by citation the paper of Ganatra and colleagues: 

[5] Ganatra B, Gerdts C, Rossier C, Johnson BR Jr, Tunçalp Ö, Assifi A, et al. Global, regional, and subregional classification of abortions by safety, 2010-14: estimates from a Bayesian hierarchical model. Lancet. 2017 Nov 25;390(10110):2372–81. 

We have added an example of public health improvements in two countries (South Africa and Nepal), and also we included a sentence that ratifies that legalization implies a commitment by the State to guarantee safe practices: 

lines 113-116: 

The law is expected to improve public health, as laws legalizing abortion create enabling environments for skilled providers and the access to safe methods [4] [5]. For example, in South Africa and Nepal, maternal mortality declined after the liberalization of abortion law [6]. In addition, legalization implies recognition on the part of the State its of obligations towards pregnant people as rights-bearers vis-à-vis the right to safe abortion and post-abortion care [7]. The passage of the new law in Argentina is consistent with international human rights treaties ratified by the country and underscores the commitment of the Argentine state to public health and sexual and reproductive rights, guaranteeing the safety and timeliness of abortions for pregnant people [8,9].

[6] Guillaume A, Rossier C. Abortion around the world. An overview of legislation, measures, trends, and consequences. Popul (English Ed INED - French Inst Demogr Stud. 2018;73: 217–306. doi:ff10.3917/pope.1802.0217ff.

 [7] Levín S. Sexual and reproductive health without freedom?: The conflict over abortion in Argentina. Salud Colect. 2018 Jul;14(3):377–89.

We move the paragraph: 

“The passage of the new law in Argentina is consistent with international human rights treaties ratified by the country and underscores the commitment of the Argentine state to public health and sexual and reproductive rights, guaranteeing the safety and timeliness of abortions for pregnant people [8,9]” to lines 117-120.

• Line 102 (Page 5): As the world abortion environment has changed dramatically since 2021 (legalization in Colombia and restrictions in the US, for example) we suggest you update this affirmation.

“Legalization of abortion in Argentina is also an important achievement that reflects broader societal change across Latin America and the Caribbean, the world region with the most legally restrictive abortion laws and policies [5].

A: Thank you for suggesting this important point. We have revised the text as follows: lines 121-122: 

Legalization of abortion in Argentina is also an important achievement that reflects broader societal change across Latin America and the Caribbean, the world region with the most legally restrictive abortion laws and policies [5]. where access to abortion has increased substantially in the past decade [10].

[10] Allotey P, Ravindran TKS, Sathivelu V. Trends in Abortion Policies in Low- and Middle-Income Countries. Annu Rev Public Health. 2021 Apr 1;42:505–18.

• Line 108 (Page 5 and in the entire paper): We suggest you use either “Pregnant people” or “women” (the latter specifying that includes trans and non-binary people) instead of using both throughout the paper.

A: Thank you for your suggestion. While we agree that it is a more parsimonious approach, for purposes of inclusivity, we believe it is important to name the fact that abortion is an issue that affects all populations with the reproductive capacity for pregnancy (including cisgender women as well as people who are transgender, non-binary, gender-fluid, intersex, and gender non-conforming).This is in line with Argentine national policy regarding legal abortion and post-abortion care, as well as international standards, such as those set by WHO. At the same time, we recognize that there are important gendered power dynamics that inform the ongoing sidelining of abortion and postabortion care, making it critical to name women as an impacted population. 

Consequently, we have decided to conserve the use both terms in the manuscript, while adding the following clarification: 

lines 205-207: “Recognizing that not everyone who may need abortion care may identify as a woman, in this study we use the terms “women” and “pregnant people'' interchangeably to refer to people with the reproductive capacity for pregnancy. “

• Line 118 (Page 6): Is there any other study that can support this sentence?

“For example, a provider who erroneously believes that abortion upon request is not legal may refuse to provide care even if they would personally be willing to perform the abortion. This issue could be magnified in settings where new laws are recently implemented. It is therefore crucial that any provider authorized to perform abortion have the correct knowledge of abortion laws and policies to ensure access to the practice and also to protect sexual and reproductive rights [12].”

A: Thank you for your request, we added the study of Puri et Al, carried on in Nepal in which authors explore providers´ knowledge and abortion denial.

 [12] Puri MC, Raifman S, Khanal B, Maharjan DC, Foster DG. Providers’ perspectives on denial of abortion care in Nepal: a cross sectional study. Reprod Health [Internet]. 2018 Dec;15(1). Available from: http://dx.doi.org/10.1186/s12978-018-0619-z

• Line 121 (Page 6): Health personnel authorized to provide abortions may vary according to the legal stipulations and health system. It may be necessary to briefly mention who is included in this “provider authorized to perform an abortion” group in Argentina.

A: We clarify who are legally authorized to do abortion practices in Argentina according to the National Law 17132 (Legal practice of medicine). 

Lines 149-150: “In Argentina, all physicians are legally authorized to perform both surgical and medical induced abortions [16]”. 

[16] Ley de Ejercicio profesional de la Medicina, odontología y actividades de colaboración. 1967. (Arg) [Internet]. 31 de Enero de 1967. Available from: http://www.saij.gob.ar/17132-nacional-regimen-legal-ejercicio-medicina-odontologia-actividades-auxiliares-mismas-lns0001226-1967-01-24/123456789-0abc-defg-g62-21000scanyel?q=%28numero-norma%3A17132%20%29&o=0&f=Total%7CTipo%20de%20Documento/Legislaci%F3n%7CFecha%7COrganismo%7CPublicaci%F3n%7CTema%7CEstado%20de%20Vigencia%7CAutor%7CJurisdicci%F3n&t=1

• Line 124 (Page 6): Legal requirements to perform abortion may vary in each country and region. It may be necessary to include which are the ones that apply in Argentina, to help the reader understand the findings and conclusions.

A: Thank you for pointing out this important point. Under the new law, the only restriction (or requirement) that exists is for abortion "on request", which the law states applies to pregnancies of 14 weeks and 6 days gestational age. There are no other additional restrictions for the national law. 

We added this information:

lines 152-153: 

In addition, providers may believe that there are additional restrictions or requirements beyond those codified in domestic laws. In Argentina, for example, under the new law, no additional restrictions (beyond gestational age for abortion “on request”) are required. 

Also we edited the heading in table 3 (page 17): “Additional requirements-restrictions not stipulated by law”, which can help the reader to better understand.

• Line 194 (Page 9): As mentioned in the section above, when you say “abortion practices” are you referring to medical abortion, surgical abortion, or both? 

A: We are referring to both practices both in this line and throughout the paper.

We clarify this in lines 253-254: 

Eligible participants were the cadres legally authorized to perform abortion practices (both medical and surgical) participating health facilities (specialists in Ob/Gyn and General Practitioners) that provided services of sexual and reproductive health. Facilities’ managers provided a list of eligible participants based on the payroll.

Please clarify here and on the entire paper.

• Line 198 (Page 9): Please specify if there are other exclusion criteria.

A: The only exclusion criterion was that eligible professionals (gynecologists/obstetricians and general practitioners) refused to give their consent. 

Lines 258-259: We added the sentence “No other exclusion criteria were applied”. 

• Line 283 (12): When addressing the years of experience, do you also have information about the specific practice they have experience in? Would you consider that a relevant element for your analysis and their willingness to provide abortion? (For example, General practitioner vs. OBYGN)

A: Thank you for this comment, in fact we only asked about years of experience and whether the participants had a specialty. We did not ask about specific practices although the centers selected to participate in the study were health centers offering sexual and reproductive health services where participants were exposed to the practice of abortion. 

Given the small number of general practitioners (n=3) and the 18 cases that refused to answer the question about what their medical specialty was, it was not possible to compare the willingness to perform abortions between both types of professionals.

• Table 2 (Page 15): How are “Personal or religious or moral reason” and “Facility reason: religious” different? Considering that you mentioned that in Argentina the conscientious objection couldn’t be institutional.

A: It is a very good point that deserves more explanation. The two concepts express different meanings and as we mention in line 175-176 page 7, conscientious objection in Argentina is an individual right only and it cannot be exercised at the institutional level. However, it made sense to ask it in the survey because there were institutions that tried to impose institutional CO, which was finally not allowed by the new law and the annexed protocol.

• Line 335 (Page 18): Figure 1 is not in the body of the paper.

A: We have reviewed the journal's requirements for figures: https://journals.plos.org/plosone/s/submission-guidelines#loc-figures-and-tables

They explicitly ask to not include figures in the main manuscript file. Each figure must be prepared and submitted as an individual file.

Also we consulted via email with the staff of PlosOne, and they confirmed this information. Please see the uploaded file “figure 1.tiff”.

• Line 345 (Page 18): When hypothesizing about the relationship between the province and willingness to provide service, don’t you consider it relevant to include the province’s characteristics that may explain better those potential connections? This could also help create specific recommendations for future actions.

A: As we mentioned before, we have added information about provinces ´characteristics in lines 282-292.

• Line 425 (Page 21): We suggest editing the sentence to show that these findings can inform potential actions instead of assuming the results suggest actions just as they are.

A: We have edited the sentence so that it reflects that the results are informative and they suggest potential actions. 

Line 525:

These findings suggest several can inform potential actions that might may improve the implementation of the law and associated guidelines, not only in Argentina, but globally. 

• Line 485 (Page 23): Can this – the survey being conducted immediately after passage of the new law – also be a limitation of your study?

A: Thank you, this is a point that really deserves to be raised. Regarding this, we added two notes indicating that the recent passage of the law may have had an impact on the degree of knowledge and exposure of professionals to the new law.

Lines 598-604: Providers may have been particularly unwilling to participate in the survey because of its temporal proximity to the change in the law; the multi-year campaign leading up to the change was heavily politicized.

Lines 619-620: First, this survey was conducted immediately after passage of the new law, and as such our results can be useful as a baseline for future comparisons, understanding that the short exposure time may have had an impact on the degree of awareness of the changes introduced.

Reviewer #3: Thank for you the opportunity to review "Factors associated with provider unwillingness to perform induced abortion in Argentina: A cross-sectional study in four provinces following the legalization of abortion on request." This is a very useful study and valuable contribution to the literature. Overall the article is well structured and results fairly well situated within Argentina’s broader socio-legal context leading to the change in legal status. A few recommendations are suggested prior to publication.

Introduction

Relevance of context and social movements for legalization is critical, however not well explained. An additional sentence or two pertaining to grassroots mobilization for “social and cultural change” would benefit readers who are unfamiliar with the national context (line 98-99).

A: Thank you for this very good observation, we proceed to add a sentences about the context of the mobilization:

lines 97-111: “The new law passed was the result of a long process that began to take shape many years earlier, led by feminist activists, networks of abortion providers, and community groups that helped people self-manage abortions [2]. Feminist movements inspired women of different generations to march massively through the streets of the National Congress wearing green insignias (the color that represents support for legal and safe abortion) creating a "green wave". These mass movements were complemented by an extensive and broadcasted debate of the bill on Access to the Voluntary Interruption of Pregnancy in the Parliament, to produce a social and cultural change that gave support to legalization [3,4].” 

[2] Ramos S, Keefe-Oates B, Romero M, Ramon Michel A, Krause M, Gerdts C, et al. Step by step in Argentina: Putting abortion rights into practice. Int J Womens Health. 2023 Jul 11;15:1003–15.

[3] Dvoskin G. Between the urgent and the emerging: Representations on sex education in the debate for abortion legalization in Argentina. Front Sociol. 2021 Jun 7;6:635137.

[4] Ramos S, Romero M, Ramón Michel A, Tiseyra MV, Vila Ortiz M. Experiencias y obstáculos que enfrentan las mujeres en el acceso al aborto [Internet]. Centro de estudios de estado y sociedad. CEDES. Buenos Aires. Argentina. 2020 [cited 2022 Jun 15]. Available from: http://repositorio.cedes.org/handle/123456789/4580

Line 120 – I appreciate the study is limited to legally authorized providers, but other health system actors’ knowledge of abortion laws and policies have implications. Data or interrogation about ways other health system actors can enable/constrain access to abortion services within the Argentinian context would be beneficial here; especially since equitable labour distribution is expanded on and recommended in the discussion.

A: This is a valuable observation. In this study we surveyed those who effectively could perform the practice (both surgical and/or medical) under current legislation (clarification: in Argentina only physicians can prescribe abortifacient drugs and perform surgical practices (cite#15: Ley de Ejercicio profesional de la Medicina, odontología y actividades de colaboración. 1967). Unfortunately we have not included other health agents and we agree with you that it would have been very good, and we will think about a future study that addresses this issue. For this study other health professionals are out of scope

Data collection

Data collection protocols are clear and detailed. Confirming that recruitment only involved a one-time email sent to potential participants? Any follow-up efforts for non-respondents?

A: We have made great efforts to get all potentially enrolled participants to respond to our study survey and we appreciate the opportunity to make this clear. We have added that information to the manuscript: 

Lines 269-271: “A weekly reminder was automatically sent by e-mail from the data center to participants who had not responded to the consent form and/or completed the survey during the study period”.

Findings

The findings are presented effectively and well-support by the accessible tables and figures.

Participants refusal to respond to whether abortion on request is a legal ground (10.3%) and willingness to provide abortion to save a woman’s life (6%) are interesting findings. Curious whether non-response to the latter questions is also associated with province, facility level and age variables in any way? If authors have analysis or inferences to draw from this (or situate it within the broader literature) it would be interesting to unpack in the discussion.

A: Thank you for highlighting this very interesting point but given the low number of cases (9 cases refused to respond to whether abortion on request and 5 cases willingness to provide abortion to save a woman's life), it was not possible to investigate in the subgroups. We certainly agree that it is something very valuable to continue researching in the future with a sample that allows this type of analysis. 

Discussion

Line 386 – Generally, but also specifically because authors highlight how the results differ from prior work on provider legal knowledge, the contemporary socio-legal context of Argentina needs to be fleshed out further. The influence and impact of the Latin American Green Wave and Causa Justa movement cannot be understated at both regional and country level. Expounding here on its relevance to the findings may better support the author’s comparison between the 2014 and 2023 studies.

A: We have added some lines that we believe meet the objective of highlighting the role of feminist movements in supporting the decriminalization of abortion in Argentina and in the Region: 

lines 474-479: 

This change in providers´ knowledge related to legislation may be due to strong recent exposure to the debate on the decriminalization of abortion in society and therefore within health institutions. Undoubtedly, this intense exposure in the media and in society is a result of the impact and influence of feminist movements, not only in Argentina but also in the Region, expanding a truly Latin American green wave which after many years of activism and massive mobilizations, managed to put the issue of abortion decriminalization on the political and social agenda.

Line 406 – Regarding analysis on clinical guidelines and SOPS, any implications here for steps to operationalize WHO’s updated Abortion Care Guidance, which recommends against grounds-based approaches in law and policy?

A: Thank you for raising this point. We think our findings serve best as an empiric example of WHO’s concerns, and have added some text in the manuscript to that effect.

 Lines 497-508: “These findings are consistent with the World Health Organization’s (WHO) observations that the interpretation of legislations based on gestational age and grounds may lead problems related to differences in interpretation among providers [30]. These differences in interpretation can lead to errors in ascertaining the eligibility of individuals seeking abortions [31] and also in difficulties for operationalizing the implementation of WHO’s new global guidelines.”

[30] World Health Organization. Geneva. WHO Abortion care guideline [Internet]. World Health Organization. 2022 [cited 2022 Jul 7]. Available from: https://www.who.int/publications/i/item/9789240039483

[31] de Londras F, Cleeve A, Rodriguez MI, Lavelanet AF. The impact of ‘grounds’ on abortion-related outcomes: a synthesis of legal and health evidence. BMC Public Health [Internet]. 2022 Dec;22(1). Available from: http://dx.doi.org/10.1186/s12889-022-13247-0

Operationalization of the new guidelines will depend on national contexts, and countries may need to take varied paths toward implementing WHO’s guidance. For example, in some places, this might be established through updated national clinical/legal guidance, while in others it may require new legislation or jurisprudence. Speculating on specific paths to operationalizing the guidelines is unfortunately beyond the scope of the present manuscript.

Given the focus on CO in the findings, I wondered if authors considered implementation strategies beyond checklist/widget approaches to improve accurate application of the law (line 428)? Recommendations for (continuing) medical education, curricula development, values clarification and attitudes transformation modalities, etc. may be particularly relevant here.

 A: We mentioned the suggested items between the lines mentioned above:

 lines 528-529: Implementation strategies should be developed that include provider support through training, continuing medical education, values clarification interventions [32], and implementation tools (e.g., checklists, screening tools) that help the provider to correctly apply requirements or restrictions, especially in contexts where there has been recent legalization and/or where there are numerous requirements and misinterpretations of the law or omissions could lead to adverse results”.

 [32] Turner KL, Pearson E, George A, Andersen KL. Values clarification workshops to improve abortion knowledge, attitudes and intentions: a pre-post assessment in 12 countries. Reprod Health. 2018 Mar 5;15(1):40.

Line 469 – adding a sub-heading for strengths/limitations may be useful for flow.

A: Thank you for your suggestion. We have added sub-headings for both items you mentioned. (Lines 589 & 616)

Thank you for the opportunity to review this manuscript. I believe the PLOS ONE editors are well positioned to ensure the above comments are addressed. Please note I believe in a transparent review process and have provided the authors and the journal editors with the same comments.

---

## [Decision Letter · Decision Letter 1]

13 Sep 2023

"Factors associated with provider unwillingness to perform induced abortion in Argentina: A cross-sectional study in four provinces following the legalization of abortion on request"

PONE-D-23-07008R1

Dear Dr. Vázquez,

We’re pleased to inform you that your manuscript has been judged scientifically suitable for publication and will be formally accepted for publication once it meets all outstanding technical requirements.

Kind regards,

Andrea Cioffi

Academic Editor

PLOS ONE

Additional Editor Comments (optional):

Reviewers' comments:

Reviewer's Responses to Questions

**Comments to the Author**

1. If the authors have adequately addressed your comments raised in a previous round of review and you feel that this manuscript is now acceptable for publication, you may indicate that here to bypass the “Comments to the Author” section, enter your conflict of interest statement in the “Confidential to Editor” section, and submit your "Accept" recommendation.

Reviewer #2: All comments have been addressed

Reviewer #3: All comments have been addressed

2. Is the manuscript technically sound, and do the data support the conclusions?

Reviewer #2: Yes

Reviewer #3: Yes

3. Has the statistical analysis been performed appropriately and rigorously? 

Reviewer #2: Yes

Reviewer #3: Yes

4. Have the authors made all data underlying the findings in their manuscript fully available?

Reviewer #2: Yes

Reviewer #3: Yes

5. Is the manuscript presented in an intelligible fashion and written in standard English?

Reviewer #2: Yes

Reviewer #3: Yes

6. Review Comments to the Author

Reviewer #2: Thank you for the opportunity to review the revisions submitted in response to our prior review of this manuscript. We thank the authors for their diligence in attending to our suggestions, and for the improvements they have implemented throughout the paper. We believe this version addresses our comments. The settings and methods sections that were confusing in the first revision have gained clarity in this revision.

Reviewer #3: Thank you for the invitation to review the revised manuscript. The authors have thoroughly addressed my comments and based on their revisions I recommend that the article be accepted at this stage.

7. PLOS authors have the option to publish the peer review history of their article (what does this mean?). If published, this will include your full peer review and any attached files.

Reviewer #2: **Yes: **Wendy V. Norman

Reviewer #3: No

---

## [Editor Report · Acceptance letter]

25 Sep 2023

PONE-D-23-07008R1 

Factors associated with provider unwillingness to perform induced abortion in Argentina: A cross-sectional study in four provinces following the legalization of abortion on request 

Dear Dr. Vázquez:

I'm pleased to inform you that your manuscript has been deemed suitable for publication in PLOS ONE. Congratulations! Your manuscript is now with our production department. 

Kind regards, 

on behalf of

Dr. Andrea Cioffi 

Academic Editor

PLOS ONE